# Comprehensive evaluation and characterisation of short read general-purpose structural variant calling software

Daniel L. Cameron [1,2], Leon Di Stefano [1] & Anthony T. Papenfuss [1,2,3,4,5]

In recent years, many software packages for identifying structural variants (SVs) using whole-genome sequencing data have been released. When published, a new method is commonly compared with those already available, but this tends to be selective and incomplete. The lack of comprehensive benchmarking of methods presents challenges for users in selecting methods and for developers in understanding algorithm behaviours and limitations. Here we report the comprehensive evaluation of 10 SV callers, selected following a rigorous process and spanning the breadth of detection approaches, using high-quality reference cell lines, as well as simulations. Due to the nature of available truth sets, our focus is on general-purpose rather than somatic callers. We characterise the impact on performance of event size and type, sequencing characteristics, and genomic context, and analyse the efficacy of ensemble calling and calibration of variant quality scores. Finally, we provide recommendations for both users and methods developers.

[1] Bioinformatics Division, Walter and Eliza Hall Institute of Medical Research, 1G Royal Pde, Parkville, VIC 3052, Australia. [2] Department of Medical Biology, University of Melbourne, Parkville, VIC 3010, Australia. [3] Peter MacCallum Cancer Centre, Victorian Comprehensive Cancer Centre, Melbourne, VIC 3000, Australia. [4] Sir Peter MacCallum Department of Oncology, University of Melbourne, Parkville, VIC 3010, Australia. [5] School of Mathematics and Statistics, University of Melbourne, Parkville, VIC 3010, Australia. Correspondence and requests for materials should be addressed to A.T.P. (email: papenfuss@wehi.edu.au)

Structural variants (SVs) are polymorphisms involving a segment of DNA that differs between individuals, or in cancer, between somatic and a normal sample, for example, ref. [1]. SVs are typically defined to be events that are >50 bp in size[2]. These include large insertions (including transposons), inversions, balanced or unbalanced translocations, amplifications, deletions and complex rearrangements, which do not fall neatly into any of these categories[3]. Understanding structural variation is important in the study of population diversity and disease[4,5], including cancer, for example, refs. [3,6,7] and is increasingly important in the clinic.

Detecting SVs using short read sequencing is a challenging problem, as the evidence for SVs resembles common sequencing and alignment artefacts. Typically, SVs are detected by looking for changes in read depth (RD), identifying clusters of discordantly aligned paired-end (PE) reads or split reads (SRs), constructing some form of assembly or a combination of these approaches. PE and RD approaches[8–29] can detect breakpoints, but unlike SR or assembly-based approaches cannot achieve single-nucleotide resolution. SR approaches can be further classified by whether they identify SRs based on upstream read alignment[24,29,30], partially aligned soft-clipped (SC) reads[15,21,22,26,29,31–35] or targeted re-alignment of one-end anchored read pairs in which only one read in the pair is mapped[18,20,26,36–39]. Assembly approaches perform alignment of assembled contigs to identify SVs[40], with a number of specialised SV assemblers having been developed[28,29,31,41,42] to avoid the computational burden of whole-genome de novo assembly.

Different SV callers support different study designs. Most general-purpose SV callers only analyse a single sample at a time, making them applicable to the analysis of normal DNA or cancer samples lacking matched normal samples (such as cell lines). In contrast, there are dedicated somatic SV callers designed to work only on matched tumour and normal pairs[43,44]. Some general-purpose tools also support optional somatic calling or the joint analysis of multiple-related samples, enabling somatic analyses through post processing (e.g. cortex, GRIDSS). Due to the lack of large, publicly available, high-quality somatic truth sets, we have focused on general-purpose SV callers rather than somatic mutation callers, although the computational challenges are not unrelated.

With over 40 short read-based general-purpose SV callers published since 2010 (Supplementary Table 1), there is a need for a comprehensive benchmark comparing a variety of SV calling approaches and implementations across of a range of datasets. Although a number of reviews have been published describing the theoretical advantages and weaknesses of different SV calling algorithms[45–53], comparative evaluation and benchmarking of software implementations has been limited. With the exception of small indels (≤50 bp) and CNVs, which we exclude here due to the existence of several benchmarking studies[54–60], publications incorporating more general SV benchmarking have primarily focused either on benchmarking frameworks or on new callers[61–63]. But with newly published callers invariably reporting favourable performance, it is difficult to discern whether the results of these studies are representative of robust improvements or due to the choice of validation data, the other callers selected for comparison, or over-optimisation to specific benchmarks.

To address these shortcomings, we have undertaken large-scale benchmarking across a range of SV callers that were selected using a careful process and represent different approaches, as well as using both real and simulated data. Our simulated data is designed to test the full range of parameters typically encountered in model organism re-sequencing studies. We have used multiple reference datasets with high-quality truth sets. We explore the impact of sequence context and show which repeat classes cause the most significant problems to SV callers. We examine the performance of quality scores. We also comprehensively evaluate simple ensemble calling approaches based on the intersection and union of different callers, showing only minor improvements in performance over the best tools. Based on these benchmarks, we provide a set of recommendations for developers and users of SV calling software.

## Results

**Method selection.** We first undertook a rigorous SV caller selection process. More than 40 SV callers, most published after 2010, were initially considered (Supplementary Table 1). We excluded specialised callers for matched tumour-normal pairs due to the lack of appropriate public validation data for these methods. Of these initial callers, we identified 14 that were highly cited and represented a cross-section of different detection approaches (indicated in Supplementary Table 1). Callers were run using the recommended parameters and results were obtained from the following 10 methods: BreakDancer, cortex, CREST, DELLY, GRIDSS, Hydra, LUMPY, manta, Pindel and SOCRATES (see Methods for details).

**Overall performance on well-characterised cell lines.** We next set out to determine the overall performance of callers using four cell line datasets with orthogonal validation data: NA12878, HG002, and CHM1 and CHM13 separately and merged as a synthetic diploid dataset. NA12878 is a well-studied Genome in a Bottle (GIAB) human cell line from the Ceph family, sequenced using 50× coverage 2 × 101 bp whole-genome sequencing (WGS) data[64] with 319 bp median fragment length. Variant calls were evaluated against the hg19 Parikh et al.[65] call set. Only deletion events were considered: despite NA12878 being the de facto standard sample for benchmarking and the publication of multiple reference call sets[7,65–67], a comprehensive 'gold standard' exists only for high-confidence single-nucleotide polymorphisms (SNPs), indels and homozygous reference regions[68]. HG002 is a second GIAB cell line for which a high-confidence SV call set has been produced by consensus calling using multiple sequencing technologies. This dataset uses 60× coverage WGS with a 555 bp median fragment length. Only calls falling within high-confidence regions of the HG002 truth set were evaluated. CHM1 and CHM13 are two haploid human cell lines, which have been sequenced to 40× coverage using 2 × 151 bp reads with 263 and 345 bp median fragment lengths, respectively, and for which a high-quality truth set is available based on PacBio long read sequencing[69]. The SVs found in these haploid cell lines are homozygous, leading to generally higher coverage of SVs in the short read data, as there are no reads 'lost' to heterozygosity or cellularity. This makes SV calling relatively easier than in heterozygous samples, or impure or heterogeneous cancers. Arguably, the primary determinant of this is simply coverage of the SV, as for most SV callers, the reads that support the reference allele have little to no impact on the SV calling. The abundant homozygous SVs in these cell lines still span many sequence contexts and allow meaningful testing of SV callers. CHM1 and CHM13 short read SV calls were evaluated against the hg38 Huddleston el al truth set (see Methods). These cell lines were also merged in silico to produce a synthetic diploid dataset with 80× coverage 2 × 151 bp WGS data with a 310 bp median fragment length.

Overall performance was considered using precision and the number of true positives called/recall. The ideal performance has high precision (low false discovery rate) and high recall (Fig. 1). Across the three samples, the assembly-incorporating callers, GRIDSS and manta, consistently performed well. Pindel has the

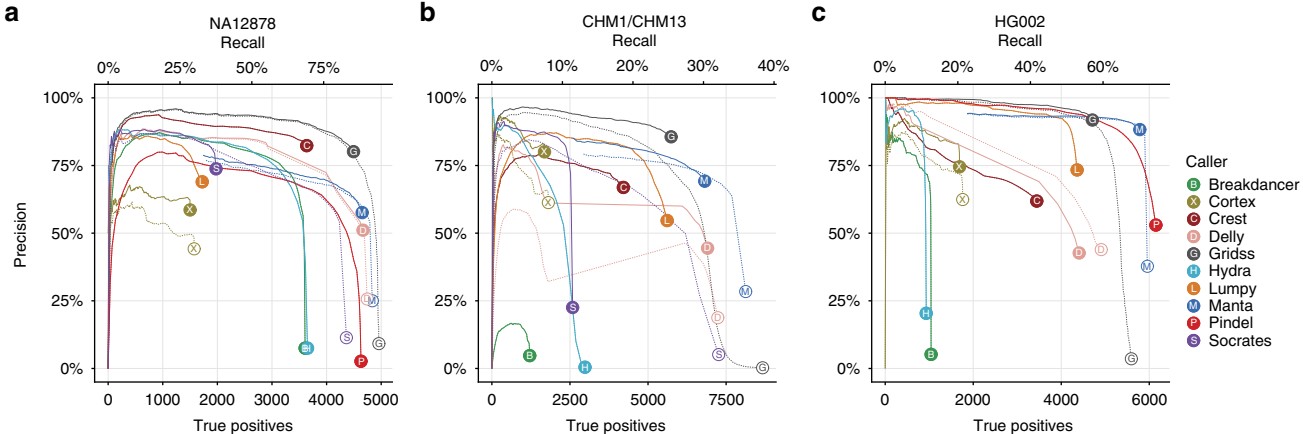

**Fig. 1** Performance varies widely among callers, but the assembly-based callers manta and GRIDSS consistently perform well. For each caller, the precision (1-false discovery rate) is plotted against the number of true positives/recall as variant quality threshold varies. The read count is used as a proxy for quality when quality score is not reported. Some callers report all calls and stratify them into pass (which can be considered as high confidence) and non-pass (low confidence), while others report only pass calls. Filled circles and solid lines correspond to calls with PASS or "." in the VCF FILTER field. Open circles and dashed lines correspond to including all calls (passed or not). The ideal caller would have a dot close with precision close to 100% and a dot as far right as possible. Each plot corresponds to a distinct human cell line and truth set: **a** NA12878: 50× coverage, 2 × 101 bp, hg19, 319 bp median fragment length; **b** synthetic diploid CHM1/CHM13: 80× coverage, 2 × 151 bp, hg38, 345 bp median fragment length; **c** HG002 60× coverage, 2 × 151 bp, hg19, 555 bp median fragment length. HG002 calls have been filtered to only regions that the truth set defines as high confidence

most variable performance: on the 2 × 101 bp reads from NA12878 (aligned to hg19), the overall precision of Pindel is very low. On HG002, the strong manta and Pindel results are driven by their high sensitivity to events under 200 bp (Supplementary Fig. 8) and their superior sensitivity on variants with long (30+ bp) micro-homology, and in low complexity regions. The sensitivity of BreakDancer and Hydra dropped dramatically on both CHM1/CHM13 and HG002 relative to NA12878 but for different reasons. These are pure PE-based methods, and thus require read alignments to span the breakpoint. On CHM1/CHM13, shorter fragments and longer read lengths reduce the likelihood of this happening and thus PE-based caller performance is degraded. On HG002, the long fragment lengths prevented the calling of shorter events and thus reduced sensitivity. The unusual behaviour of the performance curves for DELLY in CHM1/CHM13 can also be attributed to the same phenomenon, as DELLY first performs calling based on SR refinement of PE calls and only after making all such calls does it considers calls supported by SRs only. The performance of each caller on CHM1/CHM13 at 80× coverage is comparable to each of the haploid CHM1 and CHM13 results at 40× coverage (Supplementary Fig. 9). This indicates that, unlike single-nucleotide variant (SNV) calling, SV callers are robust to variant zygosity and that variant haplotype coverage is the determining factor for SV calls. The large differences in recall between the datasets can be attributed to the relative comprehensiveness of the truth sets used.

**Performance on idealised data**. We next developed a comprehensive, multi-dimensional simulation to determine how the best-case performance of each caller varied across differing event sizes, event types, and sequencing parameters. We simulated 10,000 intra-chromosomal translocations and 18,000 of each other simple SV type (heterozygous insertions, deletions, inversions and tandem duplications) up to 64,000 bp in size on chr12 (hg19). We repeated this for a range of read lengths (35, 50, 75, 100, 150, 250), fragment sizes (150, 200, 250, 300, 400, 500) and coverage levels (4×, 8×, 15×, 30×, 60× and 100×).

For the most part, simulation results recapitulate what one might expect based on the algorithmic approach of each caller

(Supplementary Figs.s 1–6). De novo assembly is required to detect large insertions, and the detection of small events requires SR analysis or assembly. The interplay between read length and fragment size for PE-based callers is complex: increasing the read length, decreasing the median fragment length and narrowing the fragment size distribution all enable the detection of events of a smaller size, but this is counteracted by a precipitous drop in ability to detect any events as paired reads start overlapping. BreakDancer is particularly problematic in this regard, as reducing fragment size from 300 to 150 bp with 2 × 100 bp reads reduces sensitivity from over 90% to <20%, while the total size of the call set remains largely unchanged. Overall, coverage above 30× generates only marginal improvements in sensitivity for non-insertion events, while a reduction in coverage below 30× reduces sensitivity. The maximal insertion sensitivity is not reached until at least 60× coverage.

Several tools do not report all event types. For example, CREST, DELLY, HYDRA and LUMPY do not detect insertions and cortex does not call inversions or tandem duplications. DELLY's poor performance on long reads/short fragments is consistent with its approach of using PEs first then SRs, but the failure to call deletions or duplications under 300 bp is hard coded into the implementation. In contrast to this, some tools are liberal in reporting certain classes of events. For example, BreakDancer, Pindel and DELLY report an inversion event even if only one of the two constituent breakpoints are present.

Several tools exhibited behaviour that appears to be unintended. Neither cortex nor CREST achieves particularly good sensitivity for any event type or size, while Pindel failed to detect 1 kb deletions and 2 kb duplications (see Supplementary Fig. 3).

**Impact of sequence context and event size on precision**. In order to determine the causes of false positives in real data, we returned to the cell line data and classified variant calls by event size, number of SNVs or indels within 50 bp of the breakpoint and sequence context using RepeatMasker[70] and tandem repeats finder (TRF)[71] annotations (Fig. 2). For deletion calls, precision peaked at 300–500 bp event size for all callers. Precision is poor for calls smaller than 100 bp, with a false discovery rate (FDR)

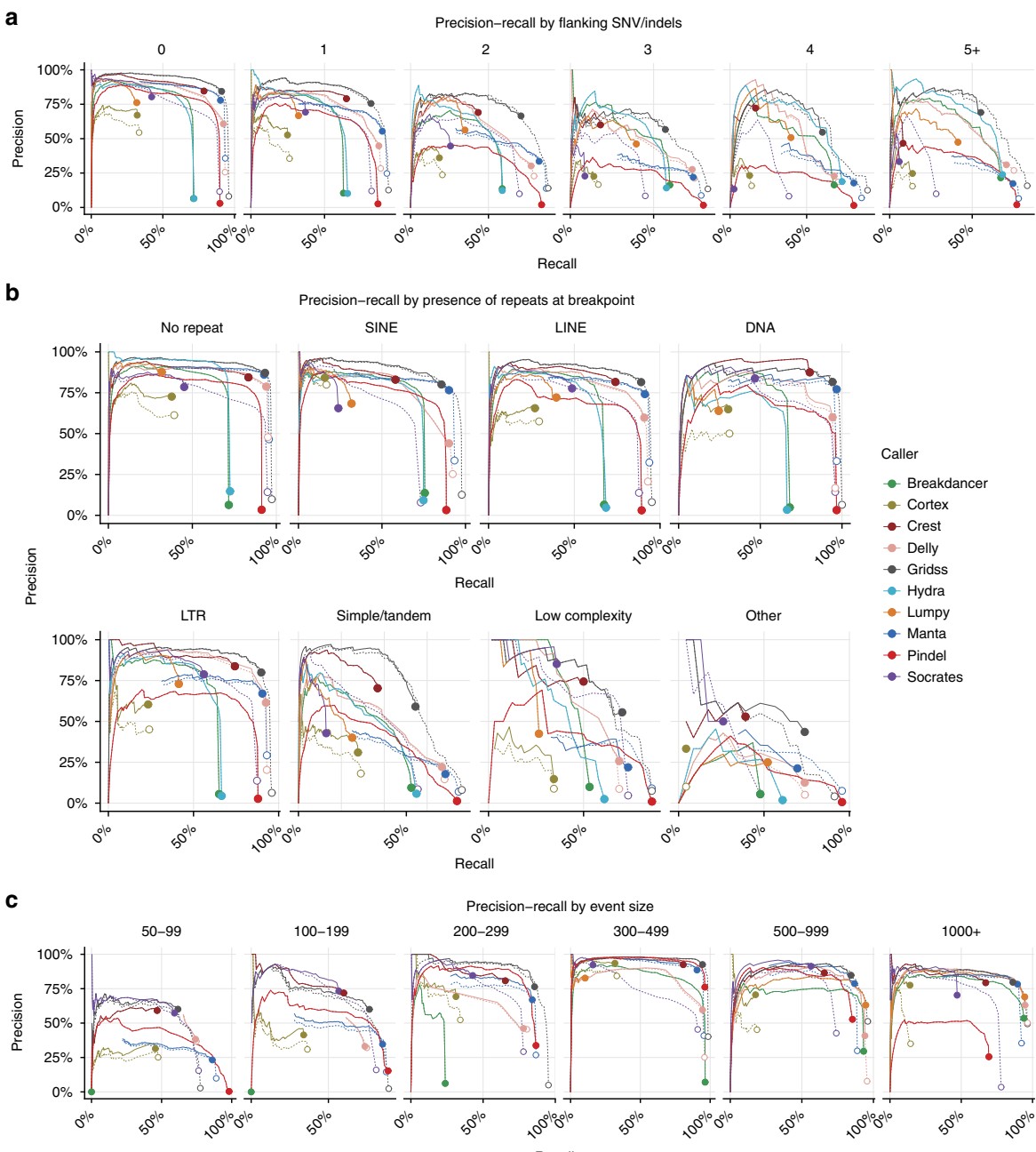

**Fig. 2** Callers perform poorly near single-nucleotide variants (SNVs) and indels, near low complexity and simple tandem repeat regions, and in detecting small events. Precision vs. the number of true positives (see Fig. 1) with calls stratified by: **a** the number of SNVs and indels within 50 bp of the variant breakpoint; **b** RepeatMasker/Tandem Repeat Finder annotation of breakpoint location. Similar repeat classes have been merged for clarity; **c** the size of the event. Open circles and dashed lines correspond to including all calls (passed or not); filled circles and solid lines correspond to calls with PASS or "." in the VCF FILTER field and are indicate of high-confidence calls

over 25% for all callers, although this is less pronounced in CHM1/CHM13 (Supplementary Fig. 7) and HG002 (Supplementary Fig. 8). Across all callers, the presence of SNVs or indels near the call breakpoint is associated with poor precision: calls with two or more small flanking variants have a considerably higher FDR than those without flanking variants. Similarly, calls occurring in low complexity, simple or short tandem repeat (STR) regions have low precision for all callers. Despite some variability, callers are generally unaffected by DNA, LINE and SINE element repeats, have an elevated FDR in LTR repeat regions and are seriously impaired only in the remaining 2% of the genome marked by RepeatMasker or TRF.

**Caller concordance and ensemble calling**. A common strategy when calling variants is to use ensembles of callers[72]. By requiring a call to be made by at least $m$ of $n$ callers (typically 2 of 3, or 2 of 4), it is thought that sensitivity or specificity can be improved compared with using any one caller in isolation. To investigate this, we analysed the overlap among the callers' true and false positives, and evaluated synthetic ensemble call sets for all possible $m$-of-$n$ rules (see Methods online).

For the NA12878 call set, most true variants are called by three or more callers, but only a small proportion of variants are called by all 10 callers (Fig. 3a). Unfortunately, identifying the remaining true variants from the large number of false positives

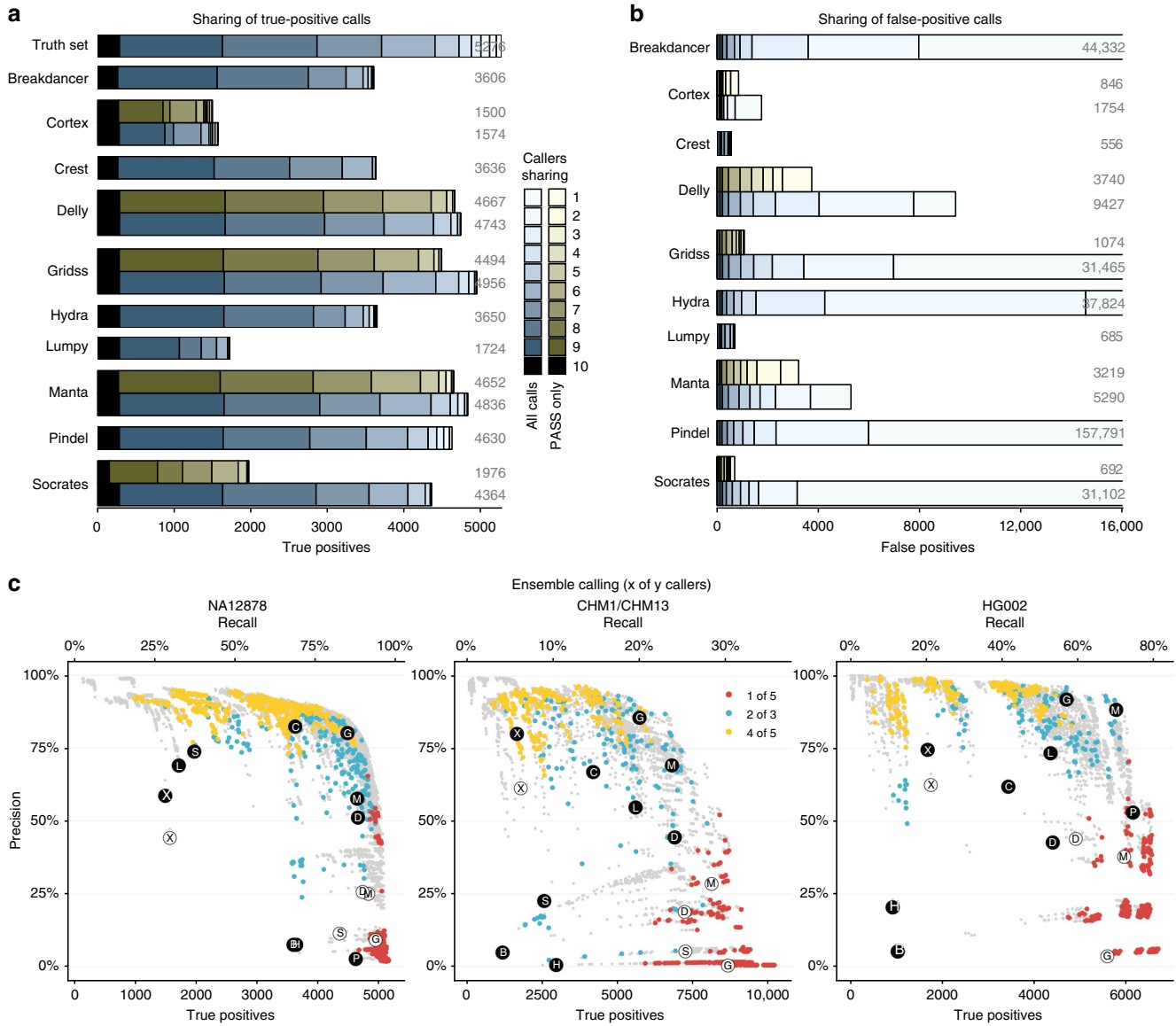

**Fig. 3** Simple ensemble-based calling does not reliably improve performance. **a** Agreement between callers in NA12878. For each caller, true-positive calls are stratified and shaded by the number of other callers in agreement. Both the full call set (blue) and the subset passing all caller-defined filters (green) are reported for callers that report filtered variants. The top bar shows the stratification of the truth set by number of callers detecting the variant. **b** Agreement between callers for false positives. **c** Precision vs. number of true positives/recall for all possible *m*-of-*n* ensembles (grey points) compared with individual callers (larger circles). 1 of 5, 2 of 3, and 4 of 5 ensembles are highlighted in colour

is difficult due to the considerable overlap among callers' false positives—four of the ten callers shared more false variant calls with at least two other callers than they made true variant calls (Fig. 3a, b). Overall, while simple ensemble calling does sometimes improve on the performance of the best individual callers (Fig. 3c), it is highly sensitive to the callers chosen for the ensemble. All ensembles that outperformed the best individual caller in any dataset (measured by *F* score) contained at least one of the assembly-based callers (manta or GRIDSS). No single ensemble consistently outperformed the best individual callers across all three datasets.

**Quality scoring**. With the total number of calls made by each caller varying by over two orders of magnitude, the choice of which calls to consider for downstream analysis is non-trivial, especially if callers do not make use of the VCF FILTER field to provide a default set of high-confidence (pass) calls. A simple but

widely used form of filtering requires that calls be supported by a certain number of reads (or read pairs), or that the caller-reported quality score be above some chosen threshold. A low threshold results in a sensitive call set at the cost of reduced precision, whereas a high threshold trades reduced sensitivity for increased precision. To determine the effectiveness of such filtering, we stratified calls by supporting read count or quality score, where available.

Greater support (or a higher quality score) generally corresponds to higher precision, with the exception of very highly supported calls, which tend to have low precision (Fig. 4). Manta is an exception to this trend, as it filters calls with very high read counts. Most callers have relatively few of these well-supported false positives, although BreakDancer does make hundreds of such calls on CHM1/CHM13 (Supplementary Fig. 10), but not on NA12878. Since HG002 calls were filtered to only the high-confidence regions of the genome included in the GIAB NIST

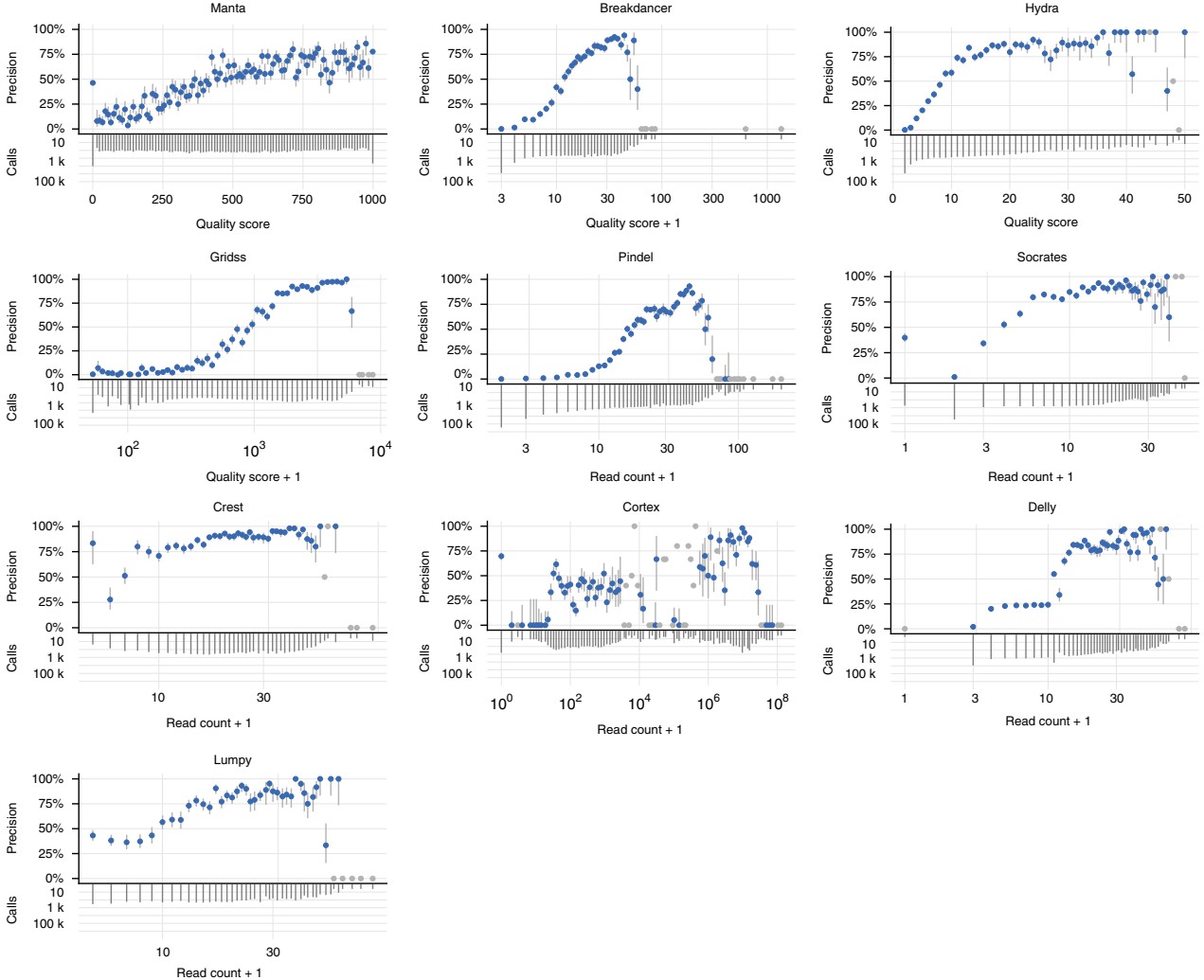

**Fig. 4** Calls with the highest read depth or quality score are often false positives. For each caller, the results for the NA12878 dataset were separated into 100 bins by either (log) read count or quality score, as indicated. For each bin, the precision (upper plot) and number of calls falling within the bin (lower plot) was calculated. Grey bars indicate 95% binomial confidence intervals for the precision. Bins with 10 or fewer calls are coloured grey and confidence interval bar omitted

truth set, the lack of precision for very high score variants exhibited in the other datasets is not present for HG002 (Supplementary Fig. 11). Of the callers reporting quality scores, only manta showed an approximately monotone behaviour across all datasets, although the majority of calls have a quality score value of 0 or 1000 (the minimum and maximum scores assigned by manta). These results further highlight the importance of considering genomic context when assessing the quality of a given SV call.

**Runtime performance**. Runtime performance of variant callers is measured for all simulated datasets (see Methods online), with execution time and CPU usage varying by over two orders of magnitude (Supplementary Fig. 12). Excluding the computationally expensive tools CREST and Pindel, single-source callers are the fastest, followed by multi-source callers, with the de novo assembly standing out as the most computationally expensive approach. In some cases, poor relative runtime performance can be attributed to the computational expense of the underlying algorithm (e.g. cortex's implementation of de novo assembly) but in the case of CREST, the poor performance is due to the reliance on external tools: the use of CAP3 assembly combined with BLAT re-alignment of soft clips means the CREST assembler runs over

an order of magnitude slower than either the Manta or GRIDSS assemblers.

## Discussion

Following a careful procedure, we selected 10 general-purpose callers that use different approaches to SV detection and benchmarked these tools. We used simulations to test the behaviour of each method on idealised data across a wide variety of sequencing parameters. Our simulations were designed to be as extensive and efficient as possible, while still avoiding interference between breakpoints calls. To our knowledge, this is the largest SV simulation undertaken and revealed surprising features in some of the callers. These simulations are a powerful tool allowing the evaluation the best-case detection capability across a range of event types and sizes. Knowing the limitations of each caller is critical to ensuring that a caller with poor detection performance on the events of interest is not used. However, simulated data is unrealistic and far too simple for accurate benchmarking, thus we have emphasised the use of reference datasets derived from cell lines with large-scale truth sets or orthogonal validation data. Our general approach is to run each caller using default parameters and visualise results using receiver operating characteristic (ROC)-like plots for each dataset. This allows exploration of the

sensitivity and precision trade-off up to the default quality score threshold, and exhaustively on unfiltered calls from those methods that also report all variants. Benchmarking on three datasets identified a few callers that consistently performed well across the multiple reference datasets.

We next explored other features of caller more carefully. We examined the impact of genomic sequence context, which highlighted that the performance of most SV callers is only marginally impacted for rearrangements in SINEs, LINEs, LTRs and DNA repeats, but is substantially impacted by simple tandem repeats and low complexity sequence. This suggests the current need to also run more specialised tools and develop approaches to integrate these calls and identifying possible future directions for improvement. We also examined the concordance of multiple callers and comprehensively assessed the utility of simple consensus-based methods. Finally, we considered the behaviour of the quality scores of each method. This revealed that calls with the highest quality scores from most methods are usually false positives. The exception is manta which demonstrated much better behaviour for these calls.

There are two main shortcomings to our general approach. First, as previously noted, due to the current lack of large-scale, high-quality reference datasets for somatic SVs, benchmarking is limited to general-purpose callers that do not require matched tumour-normal samples, although some of the methods tested do support somatic calling. For many approaches, particularly general-purpose tools that can be applied to multiple samples, the computational challenges are not unrelated and some insights may carry over. However, it is not straightforward to extrapolate to the somatic case. A separate study based on high-quality reference data from matched tumour-normal samples with large-scale validation of SVs using either an orthogonal sequencing platform or sequencing of replicate samples is needed. Second, by using default parameters, users are not able to explore calls on these reference datasets beyond the sensitivity provided by these parameters, except where the method also provides the unfiltered call set, although they can view the behaviour for more specific threshold selection in the ROC-like plots.

We have also assumed that most types of studies (e.g. population studies, cohorts, clinical genomics, $n = 1$ studies) have similar requirements, and that high sensitivity and specificity is desirable. Some users may adopt different approaches or be forced to apply tools differently by the features of their particular dataset (e.g. the fragment length). We have also assumed that most users will rely on the default parameters of a caller. For some clinical applications, the need for high specificity may outweigh the need for high sensitivity (e.g. identifying a druggable target); however, the specific disease context can make this more or less of a requirement. Outside the context of cohort studies, the ability to dig down into lower confidence calls may be desirable, so reporting all calls and scoring or labelling pass calls is an attractive feature.

Variant detection is an essential part of the analysis of genome sequencing data and sensitive, specific, usable, reliable, publicly available variant detection software is enormously valuable. Through comprehensive benchmarking of well-cited, general-purpose SV calling software spanning a range of variant detection approaches, we have shown that there is wide variation across all of these traits. To aid in the selection of callers, and the development of better SV calling software, we provide the following recommendations for users and developers:

*User recommendation: Choose a caller that uses multiple sources of evidence and assembly.* Use a recent state-of-the-art method that makes use of multiple signals of structural variation and supports single-nucleotide resolution calls (i.e. uses SR evidence). Methods that incorporate some form of assembly at breakpoints

outperform other methods. Ideally, use a method that can detect micro-homology, non-templated sequence insertions. Manta and GRIDSS are strong candidates satisfying all these criteria.

*User recommendation: Choose a caller capable of calling all relevant events.* Some SV callers are designed to detect only certain types of rearrangements and will consequently ignore or misclassify other types of events. Deletion detection is ubiquitous amongst general-purpose SV callers, but each other event type is only detectable by a subset of callers, with many callers restricted to intra-chromosomal events due to software design. GRIDSS, HYDRA, and SOCRATES address this by reporting breakpoints and leaving the interpretation to downstream analysis (e.g. CLOVE[73]). On the other hand, BreakDancer, DELLY and Pindel will report inversion events even when only one of the two breakpoints required for an inversion are present. Such misreporting makes these callers unsuitable for analysing complex events. To detect variants in regions of the genome where general-purpose SV callers perform poorly (such as micro-satellites), specialised callers such as HipSTR[74] should be used.

*User/developer recommendation: Ensemble calling is not a panacea.* Our exhaustive testing of simple ensemble callers shows that while ensembles can produce improved results compared to individual callers, no single ensemble outperforms the best individual callers on NA12878, CHM1/CHM13 and HG002. It is clear that users should not adopt simple ensemble approaches on an ad hoc basis, as an ensemble of poorly chosen callers will be easily outperformed by a single well-performing caller. The optimal ensemble requires many callers, the number and choice of which will vary between datasets, but the best ensembles include recent assembly based callers such as GRIDSS and manta. Conservative ensembles can lead to disastrously poor sensitivity, while unions of all calls can have staggeringly high false-positive rates. The main virtue of ensembles is robustness to problems with specific datasets or to errors arising in updates, including dependencies on aligner versioning.

An alternative approach to generating ensemble calls is to use assembly to integrate the results of multiple callers[75,76]. There is also the potential to integrate multiple callers using more sophisticated approaches than simple heuristic union/intersection rules, such as machine learning. These approaches may be more promising, but are not yet well developed and suffer the same inefficiency as other ensemble methods, that is, if the same evidence is being reused multiple times by each caller. Our results suggest that the incorporation of additional signals, and not merely the aggregation of multiple call sets based on the same underlying evidence, will drive performance improvements in these sophisticated ensemble callers.

For developers, we propose that developing single methods that make better use of the raw evidence, possibly in conjunction with information about genomic context, is a more efficient approach to improving sensitivity and specificity of calls.

*User recommendation: Do not use pure read-pair-based callers.* Methods relying only on PE evidence are now far from the state-of-the-art. Such callers face a significant trade-off: longer reads and shorter fragments allow the detection of smaller events, but at the cost of reduced signal strength. Critically, when reads from the same fragment start overlapping—that is, when the fragment size is less than twice the read length—the performance of PE-exclusive methods drops precipitously, as seen in Fig. 1. This behaviour is contrary to the expectation that improved sequencing technology will result in better variant calling, and there is no obvious indication of this instability in the output call set. Formalin-fixed paraffin-embedded samples are particularly susceptible to such errors, as for typical $2 \times 100$ bp the sequencing library fragment size distribution is such that the majority of read pairs overlap. The longer read lengths available from MiSeq and

NovoSeq instruments leads us to strongly recommend against using a purely PE-based caller except for the historical <100 bp read datasets they were designed for.

*User recommendation: Calls with unusually high read counts are likely to be reference genome/alignment artefacts.* For all callers (with the exception of manta), calls with very high-quality scores and supporting read counts were overwhelmingly false positives. Such calls are typically caused by reference genome alignment artefacts, and users would do well to be sceptical of high-coverage outliers, either manually examining the local alignment or filtering very high-coverage calls. Although we have used coverage as a proxy for call quality when a score is not reported by the caller, this is not ideal. While some callers did report variant quality scores, they were likely not well calibrated as probabilities.

*User/Developer recommendation: Do not consider simulation results representative of real-world performance.* Simulated results should not be considered representative of performance on real data, but rather an upper bound on actual performance. Simulations are a useful tool for debugging callers, identifying the limits of detection of algorithms on idealised data, and understanding how these limits vary with typical sequencing parameters: read length, sequencing depth, and library fragment size, but they are no substitute for extensive testing on real data.

*User/Developer recommendation: Use specialised algorithms for simple, tandem and low complexity repeats.* Although Pindel and manta are notably better than other callers at detecting small events with long homology, current general-purpose SV callers cannot yet reliably detect variants in simple or tandem repeats, and specialised algorithms are required for this. Users that require variant call information, such as microsatellite repeat expansions, in such regions should use a specialised caller. Developers should consider how to improve caller performance in simple/tandem repeat sequences, provide user guidance on caller detection limitations in these regions and provide recommendation on complementary tools.

*Developer recommendation: Benchmark using real data.* Most methods perform well on simulated data, which is useful for detecting the theoretical limits of a method or implementation, but is too simple compared to messy real data. Claims made about caller performance should be based on actual sequencing data not simulations. The NA12878 cell line is typically used as a validation set, even though the official GIAB NIST variant call set does not yet include SVs. With the availability of comprehensive SV call sets for CHM1/CHM13 and HG002, developers should include benchmarking results from these datasets when publishing new methods. Detailed manual investigation of false-positive and false-negative calls in the benchmarking samples should be used to refine the variant caller and improve performance.

*Developer recommendation: Beware of incomplete truth sets, overfitting methods to a single dataset, or training and testing methods on the same variant set.* Some methods achieved excellent performance on only a subset of the datasets. There are many possible explanations of this, but one alarming possibility is that methods are overfitted to one or few reference datasets. This may particularly be an issue with older or potentially incomplete reference datasets, such as NA12878. As more comprehensive truth sets become available, it is imperative that developers benchmark new tools against multiple datasets to ensure that certain classes of events are not overlooked. Developers should benchmark their tools to improve variant calling, but it is important to establish separate training and testing datasets to avoid the problem of overfitting. Tools should be tested on multiple diverse datasets using the most comprehensive truth sets available.

*Developer recommendation: Implement sanity checks for invalid inputs.* Implement sanity checks for inputs to prevent egregious misuse of software on inappropriate data. For PE-based callers, refusing to process sequencing data with overlapping reads, as is done by GASV-Pro and CLEVER, is a better outcome than reporting incorrect results. Similarly, checking the inferred fragment size distribution and read-pair orientation will prevent a tool designed for read-pair sequencing from reporting nonsensical results when run on a mate-pair library.

*Developer recommendation: Usability matters!* Usability is an important consideration for publicly released tools designed for general usage. Although the command-line execution of most programmes is relatively straightforward (1–2 commands, with an average of seven parameters), this is not universal. Avoiding unnecessary parameters (e.g. library fragment size distribution, which can be computed from the input file), providing meaning defaults and useful error messages, and having up-to-date documentation and a user guide not only improve the user experience but also significantly reduce the developer support burden.

*Developer recommendation: Use standard file formats such as SAM and VCF.* Using standard file formats such as SAM/BAM for input of aligned reads and VCF/BCF for variant calls. Adoption of standardised formats allows users' to more easily replace an existing tool with your software, simplifies the downstream analysis of the called variants and reduces the likelihood of users' misinterpreting the meaning of fields in the custom format (such as misinterpreting the genomic interval reported by BreakDancer as the breakpoint interval). SAM/BAM, VCF and fasta libraries are available in many languages and allow your programme to support such formats with minimal effort.

*Developer recommendation: Follow software development best practices.* All software, including bioinformatics software, can benefit from following software development best practices. Although many best practices are language-specific, others such as using version control system, unit testing, continuous integration, documentation and user guides, and explicit versioning of software releases are universally applicable.

*Developer recommendation: Report non-templated sequence insertions and micro-homologies.* Methods using only coverage or read-pair evidence cannot produce variant calls with base pair-level accuracy. While SR- and assembly-based methods are in theory capable of base pair-accurate calling, many callers do not detect and report non-templated sequence insertions. This is a problem for the clinically relevant task of determining the functional impact of a variant: knowing the full nucleotide sequence across the breakpoint is critical to correctly reconstructing amino acid-level changes. Similarly, reporting breakpoint micro-homology allows downstream tools to account for the inherent ambiguity in the breakpoint position.

*Developer recommendation: Use all available evidence, including assembly.* Unsurprisingly, our benchmarking has demonstrated that SV callers that utilise multiple signals outperform calls based on a single signal. Incorporation of some form assembly is particularly important as it allows weaker evidence, such as SC reads that are not able to be split-read aligned, to be incorporated in variant calling. We predict that incorporating RD information in an existing SR/PE/assembly-based method will lead to further improvements.

*Developer recommendation: Produce meaningful and well-behaved quality scores.* Although common for SNV and small indel callers, we are not aware of any SV callers that reports a quality score that adheres to the VCF file format definition of the phred-scaled likelihood of the variant being true. The lack of calibrated quality scores is problematic for users as without first performing a benchmarking comparison such as this, it is very difficult to determine what threshold use. By reporting a sensitive

call set with meaningful quality scores, the user is given a meaningful choice as to the relative undesirability of Type I and Type II errors. Note that such scores do not necessarily have to be incorporated into the variant caller itself. For sufficiently well-annotated variant calls, quality score calculation and calibration can be performed independently using the 'raw' variant calls as an input.

For tool developers, there is still room for improvement of SV detection from short read sequence using more sophisticated approaches, as well as opportunities to incorporate long read or linked read data. Unfortunately, the development of such tools is hampered by the scarcity of comprehensive benchmarking SV call sets. This scarcity is particularly acute for non-deletion and somatic events. To overcome this, we need more benchmarking datasets, particularly for somatic analysis, and spanning multiple rearrangement types.

For users, the number of SV callers to choose from will only continue to grow. Our benchmarking indicates that for human studies with typical sequencing parameters, manta and GRIDSS or more recent callers that include assembly with all other forms of evidence are likely to be the best-performing caller.

## Methods

**Selection of SV callers**. The choice of SV callers chosen is critical to the evaluation process. To ensure tools were evaluated based on their intended usage, only general-purpose DNA-sequencing SV callers were considered, with specialised callers such as transposable element, viral insertion, SNV/small indel or RNA-sequencing gene fusion callers excluded from consideration. To ensure that popular tools were included and that the callers selected were a representative cross-section of the algorithmic approaches used for SV detection, multiple criteria were used for the selection of callers. First, Web of Science citation counts were used as a proxy for tool popularity. Citation counts were retrieved for each tool and variant callers were included if amongst the 20% most highly cited, or amongst the 20% with the highest yearly citation rate. Second, the algorithmic approach of each caller was determined (Supplementary Table 1) and the tool with the highest yearly citation rate for each approach was added to the list. For tools with multiple publications such as GASV/GASV-Pro, VariationHunter/VariationHunter-CR/VariationHunter-CommonLAW and HYDRA/HYDRA-Multi, only the most recently published version was selected.

Using these criteria, the following SV callers were selected: VariationHunter-CommonLAW 0.0.4, GASV-Pro 20140228, Pindel 0.2.5b6, BreakDancer 1.3.5, HYDRA-Multi 0.5.2, CREST 0.0.1, DELLY 0.6.8, cortex 1.0.5.14, SOCRATES 1.13, LUMPY 0.2.11, CLEVER 2.0rc3, GRIDSS 0.11.5, SOAPsv and manta 0.29.6. Although highly cited, BreakPointer was excluded, as dRanger, the upstream calling software required for usage of BreakPointer was not publicly available.

**Variant calling pipeline**. For each dataset, the paired FASTQ input files were aligned to the relevant human reference genome (hg19 or hg38) using bwa mem (versions 0.6.1 (NA12878), 0.7.10 (simulations), 0.7.13 (HG002, CHM1, CHM13)), bowtie2 (version 2.3.2), and mrFast (version 2.6.0.1). BAMs sorted by chromosome and read name were generated for each output BAM.

For each variant caller and dataset, a driver script was generated programmatically. Using the input format required by the variant caller (paired FASTQ, chromosome sorted BAM, read name sorted BAM, or mrFast DIVET), variants were called based on the recommended settings. Recommended settings were taken from the usage message received when executing the programme with invalid arguments, the user guide, README, software website or publication in that order of preference. For variant callers that do not output VCF files, a python conversion script was created to convert the output format to VCFv4.2. Runtime performance was measured using the Unix *time* command on the variant calling shell script. Scripts were executed on a dual socket Xeon E5-2690v4 with 512GB of memory.

In line with the expected usage, software authors were not contacted directly regarding recommended settings. If a fatal error was encountered prohibiting the successful execution of the software, a message was posted to a publicly usable mailing list and an issue was raised on a publicly usable issue registry. Read-only issue registries (such as those hosted on Google Code and not migrated to Github) were not considered publicly usable. For each programme, a professional software engineer was allocated 2 days to generate the variant calling script and create any reference files required by that programme. If after 2 days a working script could not be created, the programme was deemed to have insufficient usability for widespread usage and was excluded.

VCF variant calls were converted to sets of breakpoint calls using the R StructuralVariantAnnotation package (http://github.com/PapenfussLab/StructuralVariantAnnotation) and matching was performed on the constituent

breakpoints. Breakpoints were considered matching if both the start and the end breakpoint position were within 200 bp of the true positions, and the event sizes differed by at most 25%. These criteria were selected to ensure that PE callers were not penalised for reporting the position inaccurately (particularly BreakDancer), and that our evaluation did not report spurious false positives due to unrelated overlapping events (e.g. 400 bp deletion call would not be matched with a 50 bp deletion). The two breakpoints composing inversion events were matched independently. When the called variant position was ambiguous due to micro-homology or inexact calling by PE-based callers, the breakpoint was considered matched if any position in the interval of ambiguity was within the 200 bp window. As BreakDancer does not report the breakpoint orientation of variant calls, breakpoints were considered to match even if their orientations did not. Variant calls written to the primary output file with '.' or 'PASS' in VCF FILTER column were included in the 'PASS-Only' call set. Note that this definition of a matching call is not commutative. The CHM1/CHM13 synthetic diploid truth set was constructed by taking the union of the CHM1 and CHM13 truth sets. Homozygous/duplicate calls in the truth set were handled by calculating the overlap between truth set calls using the above-matching logic and adjusting the total used to calculate recall accordingly.

Inter-chromosomal variants and variants under 50 bp were excluded from analysis. For the hg19 NA12878 dataset, breakpoints with either breakend within 200 bp any of the intervals listed in the ENCODE DAC blacklist were excluded from analysis.

Nearby SNVs and indel were calculated with bcftools version 1.3.1 using the -m multi-allelic bcftools call command-line option. No SNV or indel filtering was performed. SNVs/indels were considered near an SV breakpoint if the SNV/indel position was within 50 bp of the nominal SV position. Sequence micro-homology and imprecise breakpoint intervals were not considered when determining the number of nearby SNVs/indels.

Breakends were annotated according to their top-level RepeatMasker repeat class, with all classes other than DNA, LINE, SINE, LTR, Low_complexity and Simple_repeat collapsed into Other. Breakends in Simple_repeat regions or overlapping a TRF region (using "HG19 − 2,3,5,50 v2 Full Genome_repeats.bed" or "Homo sapiens HG38 (2,5,7,50, centr. excluded) Full Genome_repeats.bed" from the tandem repeats database http://tandem.bu.edu/cgi-bin/trdb/trdb.exe), but not a RepeatMasker region were labelled as Simple/Tandem.

For true-positive events, the variant length was considered to be the length of the variant in the truth set, and for false positives, the variant length was considered to be the length reported by the caller. As HYDRA reports all event as >500 bp, using the true variant length resulted in an apparently perfect specificity for <500 bp events, but using the variant-reported length resulted in an apparent sensitivity in to 500–1000 bp range greater than that of any other caller. HYDRA results were excluded from the final panel of Fig. 2 to prevent any confusion or misrepresentation caused by inaccurate reporting of variant length by HYDRA.

**Cell line evaluation**. For the Coriell Cell Repository NA12878 reference cell line, 50× coverage PCR-free 2 × 101 bp WGS reads from a HiSeq2000 were obtained from Illumina Platinum Genomes projects (https://basespace.illumina.com/s/Tty7T2ppH3Tr).

CHM1 and CHM13 cell line 40× coverage 2 × 100 bp Illumina WGS reads were obtained from the ENA short read archive (ENA accessions ERR1341794, ERR1341795), and the associated truth VCFs from http://eichlerlab.gs.washington.edu/publications/Huddleston2016/structural_variants. The synthetic diploid CHM1/CHM13 dataset was generated by merging the sequencing data from the individual cell lines. The synthetic diploid truth set was obtained by merging the individual truth sets. Callers were run on the 80× coverage merged data. CHM1/CHM13 variants were considered true positives if a match could be found in either the CHM1 or CHM13 truth set. For consistency with the truth sets used, hg19 was used as the reference genome for NA12878, and hg38 was used for CHM1 and CHM13. Pindel CHM1/CHM13 synthetic diploid results were excluded due to Pindel hanging on chr2. Attaching gdb to the running process indicates it was executing an $O(n^2)$ nested for loop with n over 3,000,000.

The hg19 HG002 truth set was obtained from ftp://ftp-trace.ncbi.nlm.nih.gov/giab/ftp/data/AshkenazimTrio/analysis/NIST_SVs_Integration_v0.6/HG002_SVs_Tier1_v0.6.vcf.gz with sequencing data obtained from ftp://ftp-trace.ncbi.nlm.nih.gov/giab/ftp/data/AshkenazimTrio/HG002_NA24385_son/NIST_HiSeq_HG002_Homogeneity-10953946/NHGRI_Illumina300X_AJtrio_novoalign_bams/HG002.hs37d5.60X.1.bam. Variant calls with either breakend falling outside the high-confidence Tier 1 regions defined in ftp://ftp-trace.ncbi.nlm.nih.gov/giab/ftp/data/AshkenazimTrio/analysis/NIST_SVs_Integration_v0.6/HG002_SVs_Tier1_v0.6.bed were excluded from analysis.

Ensembles call sets were generated for all combinations of callers and all *n*-of-*m* inclusion rules. The call set for each caller was compared to the truth set and to each other caller using the call matching logic previously outlined, giving a call overlap matrix. Since the matching logic allows for errors, call overlaps are symmetric but not necessarily transitive. This makes the determination of an ensemble call set ambiguous when chains of non-transitive calls are present. To overcome this ambiguity, we determined the precision and recall of the ensemble call set by first removing any call made by an ensemble caller that did not

overlapping at least *n* callers in the ensemble, then counting the remaining calls as either true positive or false positive based on their overlap with the truth set, and dividing the totals by *m*, the number of callers in the ensemble. This was performed for all calls, as well as the PASS-Only subset of call for each ensemble.

**In silico datasets.** Five variant types were simulated: insertions, deletions, inversions, tandem duplications, and unbalanced intra-chromosomal translocations. To allow precision and recall to be calculated per variant type, each dataset contained only heterozygous events of a single type. For insertion, deletion, inversion and tandem duplication events, 500 SVs of sizes 1, 2, 3, 4, 5, 6, 7, 8, 9, 10, 12, 16, 20, 24, 28, 32, 48, 64, 80, 96, 112, 128, 160, 192, 224, 256, 288, 320, 512, 1024, 2048, 4096, 8192, 16,384, 32,768, and 65,536 base pairs (bp) were inserted for a total of 18,000 SVs of each type. SVs were placed in order of the lowest coordinate genomic position with at least 2500 bp separation from any other event or ambiguous reference base (*N*). Translocations were simulated by fragmenting chr12 into 2500 bp fragments and randomly reassembling 10,000 of the resulting fragments. Variants under 50 bp were ignored.

Since not all variant callers are capable of the detection of inter-chromosomal events, all events were simulated on a single chromosome. Human hg19 chromosome 12 was chosen as it has close to the median chromosome size and GC content of the human genome, is rich in oncogenes, has been previously used for similar simulations, and all auxiliary reference files required by the variant callers were available for hg19.

Art 1.51[77] using a MiSeq error profile was used to generate simulated PE reads of lengths 36, 50, 75, 100, 150, and 250 bp at RDs of 4, 8, 15, 30, 60, and 100× mean coverage, from fragment sizes of 150, 200, 250, 300, 400, and 500 ± 10% base pairs. While reads were simulated exclusively from hg19 chr12, reads were aligned against the full hg19 reference genome.

When a result for a variant caller is presented without specifying an aligner, the result corresponds to the result for the most sensitive aligner for the given caller, event type, RD, read length and fragment size.

**Quality scores and read counts.** Variants were ranked according to the caller-reported quality score or, if no quality score was reported, by the total number of reads reported by the caller as supporting the variant.

Confidence intervals in Fig. 4 were calculated using the binom.confint function in R, using the exact method.

**Reporting summary.** Further information on research design is available in the Nature Research Reporting Summary linked to this article.

## Data availability
The datasets that support the findings of this study are: The NA12878 dataset is available from https://basespace.illumina.com/s/Tty7T2ppH3Tr. The CHM1 and CHM13 dataset is available from the ENA repository (accessions ERR1341794 and ERR1341795). The CHM1 and CHM13 SV dataset is available from http://eichlerlab.gs.washington.edu/publications/Huddleston2016/structural_variants. The HG002 dataset is available from NCBI (ftp://ftp-trace.ncbi.nlm.nih.gov/giab/ftp/data/AshkenazimTrio/HG002_NA24385_son/NIST_HiSeq_HG002_Homogeneity-10953946/NHGRI_Illumina300X_AJtrio_novoalign_bams/HG002.hs37d5.60x.1.bam) and the HG002 SVs are available from ftp://ftp-trace.ncbi.nlm.nih.gov/giab/ftp/data/AshkenazimTrio/analysis/NIST_SVs_Integration_v0.6/HG002_SVs_Tier1_v0.6.vcf.gz.

## Code availability
All code is available at http://github.com/PapenfussLab/sv_benchmark.

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

## Acknowledgements

A.T.P. was supported by an Australian National Health and Medical Research Council (NHMRC) Programme Grant (1054618) and Senior Research Fellowship (1116955), and by the Lorenzo and Pamela Galli Charitable Trust. D.L.C. was supported by an Australian Postgraduate Award. The research benefited by support from the Victorian State Government Operational Infrastructure Support and Australian Government NHMRC Independent Research Institute Infrastructure Support.

## Author contributions

D.L.C. and A.T.P. conceived the study. D.L.C. designed and executed the benchmarking. D.L.C. and L.D.S. performed the analysis with guidance from ATP. D.L.C. and A.T.P. wrote the manuscript with assistance from L.D.S.

## Additional information

**Competing interests:** The authors declare no competing interests.

