## [Peer Review File · Nature Communications]

Editorial Note: This manuscript has been previously reviewed at another journal that is not operating a transparent peer review scheme. This document only contains reviewer comments and rebuttal letters for versions considered at Nature Communications .

Reviewers' comments:

Reviewer #1 (Remarks to the Author):

The authors have generally done a good job responding to the reviewers' comments by clarifying the focus of the manuscript on germline SVs, adding new benchmarking on real data, and giving clear recommendations. I only have a few additional suggestions:

1. The authors currently appear to use the CHM1 and CHM13 haploid data separately. To mimic more realistically the performance on diploid genomes, the data from the 2 haploids are typically mixed together 1:1 to make a "synthetic diploid", as was recently described for small variants in doi: 10.1038/s41592-018-0054-7. Although not yet published to my knowledge, the authors of that work have also now made available benchmark SV calls to which the authors could compare, or they could use the Huddleston et al calls that they currently use.
2. Since the NA12878 benchmark is relatively old, the authors may want to use a new SV benchmark set from Genome in a Bottle for HG002/GM24385, which includes Tier 1 benchmark calls and regions to enable FP and FN assessment. This is public but not yet published at ftp://ftp-trace.ncbi.nlm.nih.gov/giab/ftp/data/AshkenazimTrio/analysis/NIST_SVs_Integration_v0.6/
3. The reason I was confused about the authors' results for DP callers is that I assumed they meant callers that use depth of coverage, which I now see they abbreviate RD. Since DP in the VCF spec means depth of coverage, I'd strongly recommend using a different abbreviation for discordant paired ends (e.g, PE).

Response to Reviewer

We thank the reviewer and editor for their helpful comments which we address in detail here. We have indicated changes made in red font in the manuscript and noted these changes in this response.

Reviewer #1 (Remarks to the Author):

The authors have generally done a good job responding to the reviewers' comments by clarifying the focus of the manuscript on germline SVs, adding new benchmarking on real data, and giving clear recommendations. I only have a few additional suggestions:

1. The authors currently appear to use the CHM1 and CHM13 haploid data separately. To mimic more realistically the performance on diploid genomes, the data from the 2 haploids are typically mixed together 1:1 to make a "synthetic diploid", as was recently described for small variants in doi: 10.1038/s41592-018-0054-7. Although not yet published to my knowledge, the authors of that work have also now made available benchmark SV calls to which the authors could compare, or they could use the Huddleston et al calls that they currently use.

2. Since the NA12878 benchmark is relatively old, the authors may want to use a new SV benchmark set from Genome in a Bottle for HG002/GM24385, which includes Tier 1 benchmark calls and regions to enable FP and FN assessment. This is public but not yet published at ftp://ftp-trace.ncbi.nlm.nih.gov/giab/ftp/data/AshkenazimTrio/analysis/NIST_SVs_Integration_v0.6/

Author response

Thank you for these suggestions. We have re-run all callers on the synthetic diploid created by merging CHM1 and CHM13 sequencing data sets, and compared these results to the truth set obtained from merging the individual truth sets from haploid datasets.

We also re-ran all callers on HG002. The analysis of HG002 has revealed a class of events not in the NA12878 truth set (which may be characterised as small deletions with large homology) for which manta and Pindel perform significantly better than the other callers. We have updated our results and recommendations accordingly.

Specific changes

- We modified Figure 1 to include the performance results for the synthetic diploid CHM1/CHM13 results.

Figure 1: Performance varies widely among callers, but the assembly-based callers manta and GRIDSS consistently perform well. For each caller, the precision (1-False Discovery Rate) is plotted against the number of true positives as variant quality threshold varies. The read count is used as a proxy for quality when quality score is not reported. Some callers report all calls and stratify them into pass (which can be considered as high confidence) and non-pass (low confidence), while others report only pass calls. Filled circles and solid lines correspond to calls with 'PASS' or '.' in the VCF FILTER field. Open circles and dashed lines correspond to including all calls (passed or not). The ideal caller would have a dot close with precision close to 100% and a dot as far right as possible. Each plot corresponds to a distinct human cell line and truth set: **a.** NA12878: 50× coverage, 2×101bp, hg19, 319bp median fragment length **b.** synthetic diploid CHM1/CHM13: 80× coverage, 2×151bp, hg38, 345bp median fragment length, **c.** HG002 60× coverage, 2×151bp, hg19, 555 median fragment length. HG002 calls have been filtered to only regions that the truth set defines as high confidence.

- Line 133: We included a discussion of results on CHM1/13 & HG002.

“Overall performance was considered using precision and the number of true positives called. The ideal performance has high precision (low false discovery rate) **while** calling as many true positives as possible (Figure 1). Across the three samples, the assembly-incorporating callers, GRIDSS and manta, consistently performed well. Pindel **had** the most variable performance: **on** the 2×101bp reads from NA12878 (aligned to hg19) the overall precision of Pindel is very low, but on the 2×151bp reads from CHM1/CHM13 (aligned to hg38) the long tail of low-quality calls is absent from the call set. **On HG002, the strong manta and Pindel results were driven by their high sensitivity to events under 200bp (Supplementary Figure 8) and their superior sensitivity on variants with long (30+bp) microhomology, and in low complexity regions.** The sensitivity of BreakDancer and Hydra dropped dramatically on **both CHM1/CHM13 and HG002 relative to NA12878 but for different reasons.** These are pure PE-based methods, and thus require read alignments to span the breakpoint. **On CHM1/CHM13, shorter fragments and longer read lengths reduce the likelihood of this happening and thus PE-based caller performance is degraded. On HG002, the long fragment lengths prevented the calling of shorter events and thus reduced sensitivity.** The unusual behaviour of the performance curves for DELLY in CHM1/CHM13 can also be attributed to the same phenomenon, as DELLY first performs calling based on SR-refinement of PE calls and only after making all

such calls does it considers calls supported by SRs only. The performance of each caller on CHM1/CHM13 at 80× coverage was comparable to each of the haploid CHM1 and CHM13 results at 40× coverage (Supplementary Figure 9). This indicates that, unlike SNV calling, SV callers are robust to variant zygosity and that variant haplotype coverage is the determining factor for SV calls.”

- We moved CHM1 and CHM13 performance results to Supplementary Figure 9.

Supplementary Figure 9: Performance of SV callers on CHM1 and CHM13 individually and on synthetic diploid (CMH1/CHM13). For each caller, the precision (1-False Discovery Rate) is plotted against the number of true positives as variant quality threshold varies. The read count is used as a proxy for quality when quality score is not reported. Some callers report all calls and stratify them into pass (which can be considered as high confidence) and non-pass (low confidence), while others report only pass calls. Filled circles and solid lines correspond to calls with ‘PASS’ or ‘.’ in the VCF FILTER field. Open circles and dashed lines correspond to including all calls (passed or not). The ideal caller would have a dot close with precision close to 100% and a dot as far right as possible.

- We included as Supplementary Figures the impact on genomic sequence context on SV calls in CHM1/CHM13 (Supplementary Figure 7) and HG002 (Supplementary Figure 8).
- Line 115: We added a description of the HG002 dataset.

“HG002 is a second GIAB cell line for which a high confidence SV call set has been produced by consensus calling using multiple sequencing technologies. This data set uses 60× coverage WGS with a 555bp median fragment length. Only calls falling within high confidence regions of the HG002 truth set were evaluated.”
- Line 139: We added reference to the strong manta and Pindel results for small events with large homology in HG002.

“On HG002, the strong manta and Pindel results were driven by their high sensitivity to events under 200bp (Supplementary Figure 8) and their superior sensitivity on variants with long (30+bp) microhomology, and in low complexity regions.”
- Line 218: We have refined observations on the challenges of ensemble calling.

“Overall, while simple ensemble calling does sometimes improve on the performance of the best individual callers (Figure 3c), it is highly sensitive to the callers chosen for the ensemble. All ensembles that outperformed the best individual caller in any data set (measured by F-score) contained at least one of the assembly-based callers (manta or

GRIDSS) (Supplementary Table 2). No single ensemble consistently outperformed the best individual callers across all three data sets.”

- Line 605: We updated the methods with the details of the synthetic diploid data set.
“The synthetic diploid CHM1/CHM13 data set was generated by merging the sequencing data from the individual cell lines. The synthetic diploid truth set was obtained by merging the individual truth sets. Callers were run on the 80× coverage merged data. CHM1/CHM13 variants were considered true positives if a match could be found in either the CHM1 or CHM13 truth set.”
- Line 612: We updated the methods with the details of HG002.
“The hg19 HG002 truth set was obtained from [ftp://ftp-trace.ncbi.nlm.nih.gov/giab/ftp/data/AshkenazimTrio/analysis/NIST SVs Integration v0.6/HG002 SVs Tier1 v0.6.vcf.gz](ftp://ftp-trace.ncbi.nlm.nih.gov/giab/ftp/data/AshkenazimTrio/analysis/NIST_SVs_Integration_v0.6/HG002_SVs_Tier1_v0.6.vcf.gz) with sequencing data obtained from [ftp://ftp-trace.ncbi.nlm.nih.gov/giab/ftp/data/AshkenazimTrio/HG002 NA24385 son/NIST HiSeq q HG002 Homogeneity-10953946/NHGRI Illumina300X AJtrio novoalign bams/HG002.hs37d5.60x.1.bam](ftp://ftp-trace.ncbi.nlm.nih.gov/giab/ftp/data/AshkenazimTrio/HG002_NA24385_son/NIST_HiSeq_HG002_Homogeneity-10953946/NHGRI_Illumina300X_AJtrio_novoalign_bams/HG002.hs37d5.60x.1.bam). Variant calls with either breakend falling outside the high confidence Tier 1 regions defined in [ftp://ftp-trace.ncbi.nlm.nih.gov/giab/ftp/data/AshkenazimTrio/analysis/NIST SVs Integration v0.6/HG002 SVs Tier1 v0.6.bed](ftp://ftp-trace.ncbi.nlm.nih.gov/giab/ftp/data/AshkenazimTrio/analysis/NIST_SVs_Integration_v0.6/HG002_SVs_Tier1_v0.6.bed) were excluded from analysis.”

We have refined several of the recommendations:

Line 342: User/developer recommendation: Ensemble calling is not a panacea

Our exhaustive testing of simple ensemble callers shows that while ensembles can produce improved results compared to individual callers, no single ensemble outperforms the best individual callers on NA12878, CHM1/CHM13 and HG002.

Line 368: User recommendation: Do not use pure read pair-based callers

... The longer read lengths available from MiSeq and NovoSeq instruments leads us to strongly recommend against using a purely PE-based caller except for the historical <100bp read data sets they were designed for.

Line 399: User/Developer recommendation: Use specialised algorithms for simple, tandem and low complexity repeats

Although Pindel and manta are notably better than other callers at detecting small events with long homology, current general-purpose SV callers cannot yet reliably detect variants in simple or tandem repeats, and specialised algorithms are required for this. Users that require variant call information, such as microsatellite repeat expansions, in such regions should use a specialised caller.

Line 409: Developer recommendation: Benchmark using real data

... With the availability of comprehensive SV call sets for CHM1/CHM13 and HG002, developers should include benchmarking results from these data sets when publishing new methods.

Line 420: *Developer recommendation: Beware of incomplete truth sets and over-fitting methods to a single dataset.*

Some methods achieved excellent performance on **only a subset of the data sets**. There are many possible explanations of this, but one alarming possibility is that methods are overfitted to reference datasets like NA12878. **As more comprehensive truth sets become available, it is imperative that developers benchmark new tools against these data sets to ensure that certain classes of events are not overlooked.** Developers should benchmark their tools to improve variant calling, but it is important to establish separate training and testing datasets to avoid the problem of overfitting. Tools should be tested on multiple diverse datasets **using the most comprehensive truth sets available.**

- We have made some minor changes to the text for clarity (indicated in red in the manuscript).

3. The reason I was confused about the authors' results for DP callers is that I assumed they meant callers that use depth of coverage, which I now see they abbreviate RD. Since DP in the VCF spec means depth of coverage, I'd strongly recommend using a different abbreviation for discordant paired ends (e.g, PE).

Author response

Thank you for this feedback. We have defined the acronym PE (line 50) and replaced "DP" with "PE" throughout the manuscript.

REVIEWERS' COMMENTS:

Reviewer #1 (Remarks to the Author):

The authors have done an excellent job responding to the referees' suggestions, particularly doing additional analyses with the new, more comprehensive CHM1/CHM13 and GIAB HG002 benchmark sets. I have only a few additional suggestions:

1. For all of the figures with "True positive" on the x-axis, I would find it more useful (and conventional) to have Recall on the x-axis, so that we know the fraction of variants discovered. You could still have a dynamic scale such that it doesn't go to 100%.

2. The authors state: "The performance of each caller on CHM1/CHM13 at 80× coverage was comparable to each of the haploid CHM1 and CHM13 results at 40× coverage (Supplementary Figure 9). This indicates that, unlike SNV calling, SV callers are robust to variant zygosity and that variant haplotype coverage is the determining factor for SV calls."

- This is surprising to me and difficult to know without making the x-axis Recall as suggested above. Could the authors make sure this conclusion holds after making the x-axis of both figures Recall?

3. The authors state: "There are many possible explanations of this, but one alarming possibility is that methods are overfitted to reference datasets like NA12878."

- This is a good point, but I'd suggest clarifying that this is mostly related to the reference dataset being old and limited to mostly deletions, rather than that it is on the genome NA12878.

Author Responses

We thank the reviewer and editors again for their feedback, which we have completely addressed. We address each point below.

REVIEWERS' COMMENTS:

Reviewer #1 (Remarks to the Author):

The authors have done an excellent job responding to the referees' suggestions, particularly doing additional analyses with the new, more comprehensive CHM1/CHM13 and GIAB HG002 benchmark sets. I have only a few additional suggestions:

1. For all of the figures with "True positive" on the x-axis, I would find it more useful (and conventional) to have Recall on the x-axis, so that we know the fraction of variants discovered. You could still have a dynamic scale such that it doesn't go to 100%.

Done. We have included Recall in all relevant figures and agree it is useful.

In Figures 1 and 3, we added it as the upper x-axis and retained the number of true positives, since it is more consistent between datasets, while recall varies significantly according to the quality and completeness of the truth set.

Line 150: Added the following text explaining the differences in recall between the data sets:

"The large differences in recall between the data sets can be attributed to the relative comprehensiveness of the truth sets used."

2. The authors state: "The performance of each caller on CHM1/CHM13 at 80× coverage was comparable to each of the haploid CHM1 and CHM13 results at 40× coverage (Supplementary Figure 9). This indicates that, unlike SNV calling, SV callers are robust to variant zygosity and that variant haplotype coverage is the determining factor for SV calls."

- This is surprising to me and difficult to know without making the x-axis Recall as suggested above. Could the authors make sure this conclusion holds after making the x-axis of both figures Recall?

Done. We have included recall in the x-axis of relevant figures, including Supp Figure 9 which supports this observation. Sample coverage has also been added to the Supp Figure 9 subheadings to make it clear we are comparing 40x haploid to 80x diploid. The conclusion that SV calling is performance is determined by haplotype coverage and not variant zygosity still holds.

3. The authors state: "There are many possible explanations of this, but one alarming possibility is that methods are overfitted to reference datasets like NA12878."

- This is a good point, but I'd suggest clarifying that this is mostly related to the reference dataset being old and limited to mostly deletions, rather than that it is on the genome NA12878.

Done. Thank you for this feedback.

p18: We have modified this recommendation as follows:

"Developer recommendation: Beware of incomplete truth sets, overfitting methods to a single dataset, or training and testing methods on the same variant set. Some methods achieved excellent

performance on only a subset of the data sets. There are many possible explanations of this, but one alarming possibility is that methods are overfitted to one or few reference datasets. This may particularly be an issue with older or potentially incomplete reference datasets, such as NA12878. As more comprehensive truth sets become available, it is imperative that developers benchmark new tools against multiple data sets to ensure that certain classes of events are not overlooked. Developers should benchmark their tools to improve variant calling, but it is important to establish separate training and testing datasets to avoid the problem of overfitting. Tools should be tested on multiple diverse datasets using the most comprehensive truth sets available.”

Other changes:

Please note that we have updated Figure 1b. We determined that Pindel and CREST had crashed on the CHM1/13 synthetic diploid data, and re-ran both. We were able to get CREST to complete successfully and its results show improvement. However, we have removed PINDEL from this panel only, as tracing showed it would never complete. We have added a note of this into the Supplementary Information.